# Terahertz Shielding Properties of Carbon Black Based Polymer Nanocomposites

**DOI:** 10.3390/ma14040835

**Published:** 2021-02-09

**Authors:** Klaudia Zeranska-Chudek, Agnieszka Siemion, Norbert Palka, Ahmed Mdarhri, Ilham Elaboudi, Christian Brosseau, Mariusz Zdrojek

**Affiliations:** 1Faculty of Physics, Warsaw University of Technology, Koszykowa 75, 00–662 Warsaw, Poland; agnieszka.siemion@pw.edu.pl (A.S.); mariusz.zdrojek@pw.edu.pl (M.Z.); 2Institute of Optoelectronics, Military University of Technology, Kaliskiego 2, 00–908 Warsaw, Poland; norbert.palka@wat.edu.pl; 3Laboratory of Sustainable Development and Health Research, Faculty of Sciences & Techniques Cadi Ayyad University, A. Khattabi BP 549, 40 000 Marrakesh, Morocco; a.mdarhri@uca.ma (A.M.); i.elaboudi@uca.ma (I.E.); 4Lab-STICC, French National Center for Scientific Research, University of Brest, 6 Avenue Le Gorgeu, CEDEX 3, 29238 Brest, France; Christian.Brosseau@univ-brest.fr

**Keywords:** carbon black, polymer composite, terahertz frequency range, electromagnetic shielding, absorption coefficient

## Abstract

The majority of industry using high-speed communication systems is shifting towards higher frequencies, namely the terahertz range, to meet demands of more effective data transfer. Due to the rising number of devices working in terahertz range, effective shielding of electromagnetic interference (EMI) is required, and thus the need for novel shielding materials to reduce the electromagnetic pollution. Here, we show a study on optical and electrical properties of a series of ethylene co-butyl acrylate/carbon black (EBA/CB) composites with various CB loading. We investigate the transmittance, reflectance, shielding efficiency, absorption coefficient, refractive index and complex dielectric permittivity of the fabricated composites. Finally, we report a material that exhibits superior shielding efficiency (SE)—80 dB at 0.9 THz (14.44 vol% CB loading, 1 mm thick)—which is one of the highest SE values among non-metallic composite materials reported in the literature thus far. Importantly, 99% of the incoming radiation is absorbed by the material, significantly increasing its applicability. The absorption coefficient (α) reaches ~100 cm^−1^ for the samples with highest CB loading. The EBA/CB composites can be used as lightweight and flexible shielding packaging materials for electronics, as passive terahertz absorbers or as radiation shields for stealth applications.

## 1. Introduction

For the last several years, the terahertz industry has experienced rapid growth, mainly due to new advancements in the main fields of terahertz technology, namely the terahertz sources, detectors and absorbers [1,2,3,4,5,6]. In this work, we concentrate on terahertz absorbers, as they play an important role in many interesting technologies, such as electronic devices packaging, electromagnetic safety shielding, stealth applications, radar systems or thermal detectors [7,8,9,10,11]. A good shielding material should exhibit at least the minimum SE that at the moment is 20 dB, which corresponds to shielding 99% of incident EM radiation [12], favorably in a broad frequency range.

Although, currently, metals and composites based on metallic nanoparticles are still the most commonly used materials for EMI shields, there is a strong alternative—polymer composites with carbon nanofillers. Polymer composites based on carbon nanomaterials combine the mechanical durability and flexibility of the polymer matrix and optical and electrical properties of highly absorptive and electrically conductive carbon fillers. The addition of carbon fillers, e.g., graphene, carbon nanotubes or carbon black, can greatly influence the shielding properties of normally transparent polymer matrixes. The specific optical and electrical properties can be tuned by the inner structure of the composite and carbon filler loading. Unlike metals, polymer composites are resistant to corrosion and can be fabricated via modern industrial methods, like molding or extrusion.

Carbon based polymer composites are widely used for EMI shielding in the microwave range, especially the frequencies including the X band (8.2–12.4 GHz) [13,14]. Zeng et al., reported a highly loaded (76 vol%) multi-walled carbon nanotube/waterborne polyurethane composite exhibiting high SE of 80 dB in the microwave X-band for a 0.8 mm thick sample [15]. There are a few examples of polymer composites employing carbon black as a nanofiller for microwave EMI shielding [16,17]. Mondal et al. showed a carbon black/chlorinated polyethylene elastomer matrix, which exhibited SE of 38.4 dB at 30 vol% CB loading and 1 mm thick sample [18]. Research on SE of polymer composites enhanced with carbon nanofillers often lacks thorough analysis of the optical properties of said materials, such as the distinction of main shielding mechanism or the investigation of the dielectric constants or absorption coefficient. It is also not uncommon for such research to deal only with the reflective loss of studied materials, which gives information only on the reflection-based shielding.

Time domain spectroscopy in terahertz range (THz-TDS) provides detailed information on the optical and dielectric properties of different materials of interest. However, currently. the literature on terahertz regime is still relatively less developed than the microwave range and not so well described. For example Chamorro-Posada et al. presented a comparative study of absorption coefficient and refractive index of polyurethane composites with graphite, needle coke, graphene oxide (GO) and reduced graphene oxide (rGO) fillers [19], showing the influence of carbon nanofiller loading on the absorption coefficient and refractive index. The highest reported absorption coefficient reached over 120 cm^−1^ for a PE/graphite composite with 2.5 wt% filler loading. Taha et al. [20] reported a thorough study of dielectric properties of epoxy composites with two types of carbon nanofillers—carbon nanofibers and multi-walled carbon nanotubes MWCNT—with regard to carbon filler loading and frequency, demonstrating reasonable SE for MWCNT filled composites, reaching a maximum SE of ~20 dB for a composite with 2.5 wt% CNT content between 0.3 and 1.0 THz. In 2007, Song et al. [21] reported polyethylene composites with CB loading ranging 1.04–13.27 vol%, exhibiting absorption coefficient of $>90\;{\mathrm{cm}}^{-1}$ at 0.4 THz for the composite with highest CB concentration; the study also mentioned the refractive index of said composites. Macutkevic et al. showed that epoxy resin composites loaded with CB particles up to 1.5% concentration exhibit transmittance similar to that of pristine epoxy [22]. Such low absorption most likely stems from low CB loading level, and, to verify this, higher CB content in the composites should be studied. To the best of our knowledge, there are no recent studies on polymer composites with high concentration of CB fillers.

Having in mind this niche in the literature, we investigate a handful of optical and dielectric properties of carbon black-based composites, including transmittance, reflectance, EMI SE and its components, absorption coefficient, refraction index, dielectric permittivity, loss tangent and resistivity with respect to the composite filler loading and frequency. The above properties are investigated in terahertz range from 0.2 to 2 THz for a series of ethylene co-butyl acrylate (EBA)/carbon black composites, which has not yet been reported in this frequency range. It is of note that the spherical CB particles are less frequently studied than other fillers like graphene or carbon nanotubes, embedded in polymeric matrix. A vast selection of filler concentration is studied—from 4.08 to 14.44 vol%—enabling proper analysis of the influence of CB particles loading on the properties of the fabricated composites. We report a series of highly shielding EBA/CB composites—SE over 20 dB throughout 64–98% of the measured range (depending on a specific CB loading), with main shielding mechanism defined as absorption. The highest measured SE was exhibited by the composite with 14.44 vol% CB loading—80 dB for 1 mm thick sample at 0.9 THz—which is one of the highest SE reported in the literature. Additionally, we show an evolution of electrical conductivity as a function of frequency. Finally, we point out that the absorption in EBA/CB composites is relative to the dielectric loss.

## 2. Methods and Materials

### 2.1. Material: EBA/CB

In this investigation, an ethylene co-butyl acrylate copolymer filled by acetylene carbon black (Denka Black) was used. The butyl acrylate monomer (4.3 mol%) contains butyl ester side groups providing a polar character and a relatively low crystallinity (20 vol%), which can be enhanced by introducing the CB particles. The crystallinity value reaches ~30% for samples containing 14.4 vol% of CB particles, as reported in our previous work [23]. EBA matrix exhibits glass transition temperature at a level of *T*_g_ ≈ −75 °C, melting temperature point at 90 °C and melting index of *T*_g_ ≈ 150 g/10 min [23,24]. A statistical transmission electron microscopy analysis on different areas of a set of images (~100 measurement points) shows that the size distribution of the spherical particles follow a Gaussian law where d ≈ 34 ± 2 nm [24]. The specific surface of said spherical particles, which strongly depends on the size of the primary particles, is 63 m^2^g^−1^. The EBA matrix was chosen due to its compatibility with CB particles, namely easy processing, good dispersion and low processing cost, with a low percolation threshold (~0.08) [24].

To fabricate EBA/CB composites, the EBA copolymer and acetylene CB particles were mixed at specified ratio (to achieve set CB loading), pelletized and extruded into tapes. Then, the tapes were press molded for 30 min at 180 °C and degassed in a vacuum oven for 24 h at 80 °C. The CB dispersion in the EBA matrix was investigated at nanometric scale by conductive probe atomic force microscopy (AFM, Nanoscope III AFM, Digital Instruments, Inc., Santa Barbara, CA, USA), which, when used in contact mode, permits a mapping of the local electrical resistance simultaneously with classical topography images. The former information is more efficient to evaluate the effective dispersion of CB through the polymer matrix regarding the high difference of the electrical resistance between the two constituents. For illustration purposes, Figure 1a shows AFM images for three samples containing 6.75, 7.97 and 12.04 vol% of CB. This picture indicates clearly that the filler is finely dispersed in the polymer matrix. Complementary information was obtained by SEM as reported elsewhere [25]. The number and size of particle clusters increase with CB content up to a point, where oversized agglomerations of CB particles start to reduce the quality of the dispersion, which occurs when CB concentrations exceeds 22 vol%. Samples prepared for this study exhibited a uniform dispersion of the filler particles in the matrix, which allowed an enhancement of interfacial properties and optimum transfer of the mechanical stress between the filler and the matrix. We fabricated nine series of samples, each with different CB volume fraction ranging from 4.08 to 14.44 vol% (specifically, 4.08, 5.10, 6.75, 7.31, 7.97, 10.86, 12.04, 13.10 and 14.44 vol%) in form of small (2 cm×2 cm) flat cuboids each 800±10% μm (as shown in Figure 1b). The densities of the samples increase linearly with CB volume fractions from 0.925 to 1.116 g cm^−1^. Figure 1c shows a SEM image of a cross section through the composite, revealing the inner structure of the composite with the spherical CB particles in the polymer matrix.

### 2.2. Characterization Methods

Measurements in terahertz range were conducted via time-domain spectroscopy, using TeraView spectrometer Spectra 3000, (Cambridge, United Kingdom). The spectrometer is based on an 800 nm femtosecond laser generating 50 fs pulses and provides data in range 0.06−4 THz ($2–120\;{\mathrm{cm}}^{-1}$). For the terahertz measurements, Rapid Scan mode was used, which employs 30 scans per second with $1.25\;{\mathrm{cm}}^{-1}$ resolution. Measurements were conducted in ambient temperature and atmosphere. Before each measurement, the sample thickness was measured. Each sample underwent a series of scans to ensure the accuracy of the collected data. The spectrometer software was used to calculate the refractive index, absorption coefficient, transmittance, reflectance and the complex dielectric function of the measured composites. For more straightforward data analysis and comparison, we normalized the data to represent the features of a 1 mm thick sample. Volume resistivity measurements were conducted with a single post dielectric resonator (from QWED, Warsaw, Poland) with an operating frequency of 5 GHz and later used for AC conductivity analysis. The details of conductivity analysis (DC to 1 GHz) were described in other work [25].

SE is comprised of three components—reflection SE_R_, absorption SE_ABS_ and multiple inner reflection SE_MIR_ components—and the following relation is true: $SE_{TOT}=SE_{R}+SE_{ABS}+SE_{MIR}$. The total SE is based on transmittance experimental data and is calculated as follows: $SE_{TOT}=-10\log_{10}T$ [dB], where T is the power ratio of transmitted signal and incident radiation. The components of SE are crucial for investigation of main shielding mechanism of the studied material. The presence of SE_MIR_ component usually decreases total SE, and it can be omitted if the wavelength of incident radiation is bigger than the thickness of a measured sample, when the SE_ABS_ of the sample is higher than 10 dB or the SE_TOT_ is higher than 15 dB [26,27,28]. In this work, the multiple inner reflection component is negligible. The other components are calculated as follows: $SE_{R}=-10\log_{10}\left(1-R\right)$ and $SE_{ABS}=-10\log_{10}\left(T/1-R\right)$. T and R are the measured transmittance and reflectance.

## 3. Results and Discussion

### 3.1. Shielding Efficiency

SE is a commonly used quantity to describe lossy materials for shielding applications. Both transmittance and reflectance can be utilized to calculate the total SE (in dB units) and its components. Figure 2 shows transmittance and reflectance spectra (in percent unit, right-hand axes) of the fabricated EBA/CB composites in the terahertz range up to 2 THz. The spectra shown in Figure 2 (top) were cut at different frequencies (0.9  to  1.8 THz depending on the CB loading), where the measurement became unreadable and unreliable. For composites with CB loading higher than 14.44 vol%, the maximum measured SE_TOT_ value is unlikely to improve, and the rise of slope of SE_TOT_ would make the measurements unreadable at lower frequencies. The calculated SE_R_ and SE_TOT_ values are displayed on the left-hand axes. The studied samples exhibit total SE of 20 dB or higher for at least 64% of the studied range (up to 98% for the highest CB loading), which means the EBA/CB composites shield over 99% of the incident radiation in more than half of the studied frequency range. Regardless of the specific CB loading in a composite, the total SE values grow almost linear with the growing frequency. Such SE frequency dependence has also been observed in other carbon related materials, such as rGO papers or polymer composites with carbon additives [19,29], however maximum achieved SE was lower. Specific CB loading can be associated with a slope value of the total SE. The higher the loading is, the faster the SE_TOT_ values grow. For example, the slope for a 4.08 vol% sample equals 31.11 dB/THz, while the sample with 14.44 vol% CB loading shows a growth rate of 89.68 dB/THz. On the other hand, the reflectance does not seem to be influenced by the CB loading. Reflectance remains under 18% in the whole studied frequency range (only 1 dB in SE_R_), decreasing quickly below 10% at ~0.5 THz. In addition, the reflectance shows no characteristic peaks and does not follow any trend regarding the CB loading or frequency. Such low reflectance (and low transmittance) is a positive trait for a terahertz absorber, as it would ensure most of the incident radiation is absorbed by the material instead of being reflected back into the environment. It is not common to find low reflection shielding materials in the literature; some works concentrate mainly on the reflection loss measurements instead of SE [4,30,31].

When studying SE, it is desirable to distinguish the main shielding mechanism contributing to the total SE. As the multiple inner reflection component was excluded in this study, we concentrate on the absorption and reflection components. Figure 3 shows the $SE_{ABS}\;$ and $\;SE_{R}$ components side by side for each type of fabricated composite at a specified frequency (0.9 THz). It is clear the total SE is dominated by the absorption component, as it makes up 99% of total SE at 0.9 THz. The reflection component exhibits no specific trend related to either the CB loading or frequency, and its mean value is 0.3±28% dB. Unlike the reflection component, SE_ABS_ depends on the CB loading. At any specific frequency, the SE_ABS_ exhibits higher values for greater CB concentrations, which we already postulated by calculating the slope values of SE_TOT_. Thus, we can state that the total SE is absorption based and dependent on the CB loading.

### 3.2. Absorption Coefficient and Refractive Index

Figure 4a shows the refractive index of fabricated composites. The refractive index shows no characteristic peaks, and its values range from 2 to over 3. The refractive index also exhibits higher values for composites with higher CB loading. These values are higher than those of pure EBA [32], but much lower than those of bulk carbon materials, such as reduced graphene oxide films [33]. The high refractive index of bulk carbon materials and influence of CB content on the values of refractive index of CB/EBA composites may suggest the composites show low refractive index, because of the EBA polymer matrix, but can be enhanced with highly refractive CB inclusions. Similar n values are reported for other polymer composites with carbon nanofillers [21,34,35].

Figure 4b shows the absorption coefficient of EBA/CB composites with different CB loading. The addition of CB particles clearly enhances not only the absorption coefficient $\left(\alpha \right)$ but also slope of $\alpha \left(\;\mathrm{Frequency}\right)$. The higher is the CB content in the composite, the higher is the α. In addition, the CB loading seems to have bigger impact at higher frequencies. The absorption coefficient values grow thrice as much with growing filler loading at 1.5 THz than they do at 0.5 THz. Absorption coefficient values reported in this work are consistent with data on similar materials found in the literature, for example a HDPE/CB composite reaches over 100 cm^−1^
α value at 15 vol% CB loading, while a polypropylene-based composite enhanced with multiwalled carbon nanotubes shows maximum absorption coefficient of ~80 cm^−1^ at 1.4 THz, which also exhibits growth trend with changing frequency [36,37].

### 3.3. Dielectric Permittivity and Loss Tangent

To better understand the influence of the filler on EMI shielding properties, it is interesting to analyze the dielectric properties of the composite—the dielectric constants and the resulting loss tangent. Figure 5 shows the frequency dependence of dielectric permittivity (real ϵ′ and imaginary ϵ″ part of the dielectric constant) and the calculated loss tangent tanδ.

Clearly, there is a strong CB loading influence on the dielectric properties. The growing CB loading strengthens both the real and imaginary parts of dielectric permittivity. In addition, the samples with higher CB concentration (over 6.75 vol%) exhibit a dependence on frequency—with the growing frequency, the ϵ′ values decrease. That trend is not visible for the samples with lower CB loadings. Such influence of filler concentration on dielectric properties has been reported in other work on carbon black and other carbon materials [22,38]. Okano et al. showed a styrene butadiene rubber composite with two carbon black weight loadings—1.5 and 3.0 wt% (0.09 and 0.19 vol%)—and pure elastomer with no carbon black additions for reference [39]. They reported a similar change of ϵ′ with the changing CB loading, and the values of ϵ′ for the 15 and 30 wt% composites correspond with the ϵ′ values of our composites with 6.75 and 13.10 vol%, respectively. Although the high CB content increases the loss tangent values, it still exhibits very low values that are comparable to those of pristine polymers [40].

The loss tangent is calculated as a relation of imaginary and real part of dielectric constant $\tan\delta =\epsilon^{\prime \prime }/\epsilon \prime$, and it represents the dielectric power loss in the material. Figure 6 shows the loss tangent in relation to the absorption coefficient of the fabricated composites. Since the THz wavelengths used in this work are much higher than the typical CB particle size (average diameter is about 32 nm [24]), the main origin of dielectric losses can be attributed to absorption. There is a clear relation between those two features, as a sample exhibiting a high absorption coefficient will also have high dielectric loss tangent. However, the specific slope of the α(tanδ) relation changes with the frequency—the higher the frequency is, the faster the absorption values grow.

### 3.4. Conductivity

Figure 7 shows the conductivity (σ_ac_) of fabricated CB composites in relation to the frequency. The general trend displays an increase in conductivity from DC to 5 GHz. This behavior agrees well with the universal dielectric response [41,42] which predicts an exponential dependence of σ_ac_ versus frequency. From that model, the dependence of the effective conductivity on the frequency is consistent with a hopping conduction mechanism. For supercolated samples, finite clusters also contribute to conduction when the frequency exceeds some critical value F_c_, which increases with volume fraction of CB [25]. Below F_c_, conductivity is independent of frequency and only infinite clusters contribute to σ_ac_.

Bychanok et al., in their work on graphite nanoplatelets (GNP), showed a comprehensive discussion on the origin of dielectric losses in their composites [38]. They suggested an alternative theory to the one presented by Chamorro-Posada et al. [19], which is based on contribution from certain vibrational modes and stacking of graphene layers. According to the alternative theory, the origin of the absorption peak is related to the geometrical and conductive properties of GNP. The increasing CB concentration should enable the formation of conduction paths that are beneficial for the SE and conductivity, however it also promotes agglomeration of conductive particles, changing the geometry of a single conducting unit. Similar to Bychanok et al., we also observed the subtle interplay of SE and conductivity, as the maximum values of the SE and the highest deviation from pristine polymer of dielectric properties occur for samples with the highest (over 10 vol%) CB loading. This is an interesting observation, as the conductivity is usually associated with high reflection based shielding mechanisms [43].

## 4. Conclusions

A study of optical properties of polymer composites enhanced with various CB loadings in terahertz range (0.2–2 THz) was conducted. We studied SE and its components, refractive index, absorption coefficient, dielectric permittivity and loss tangent of the fabricated composites. We showed strong CB loading and frequency dependence of the absorption component of SE and the absorption coefficient. We showed the CB content influences the refractive index, dielectric constants and, consequently, the loss tangent. Only the reflection component of the SE shows no visible trends frequency and CB loading wise. A broadband characterization and analysis of conductivity from DC to the GHz range of frequencies allowed us to estimate the conduction losses which underlie the absorption mechanism. Finally, we report a series of EBA/CB composites exhibiting SE over 20 dB in majority of the studied range (specific range depends on the CB loading). The 14.44 vol% EBA/CB composite reaches 80 dB at 0.9 THz, and we expect it to keep or exceed that value for higher frequencies. Importantly, the main shielding mechanism of the EBA/CB composites is absorption, which makes them suitable for stealth applications, i.e., in military sector.

## Figures and Tables

**Figure 1 materials-14-00835-f001:**
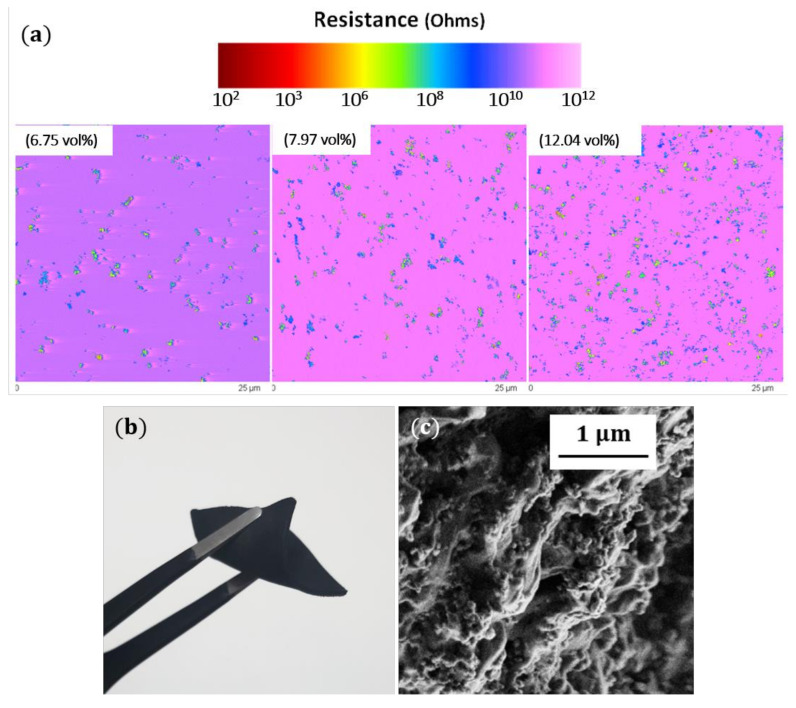
(**a**) AFM-Resiscope electrical micrographs containing three volume fractions of carbon black nanoparticles (CB). The local resistance distribution is a signature of a best dispersion of CB throughout the matrix. The size of scanned areas and the thickness of samples are, respectively, 25 × 25 µm^2^ and 3 µm. The color bar indicates the resistance in ohms. (**b**) Image of a CB/EBA composite sample. (**c**) SEM image of a cross section of the CB/EBA composite showing the spherical CB nanoparticles. Both pictures (**b**,**c**) show CB/EBA composites with 14.44 vol%.

**Figure 2 materials-14-00835-f002:**
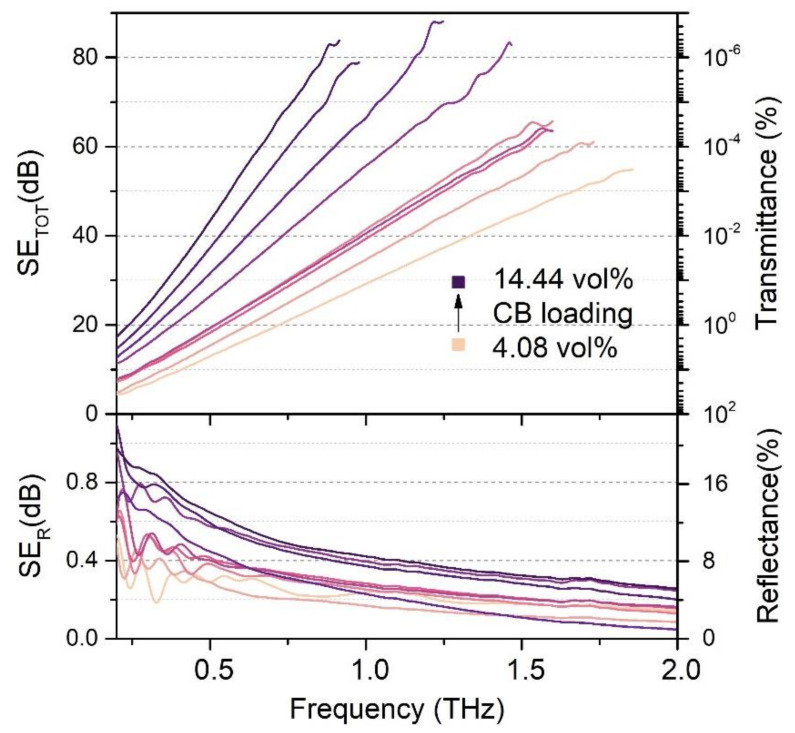
Basic optical properties of CB/EBA composites—transmittance and reflectance together with calculated total SE and its reflection component.

**Figure 3 materials-14-00835-f003:**
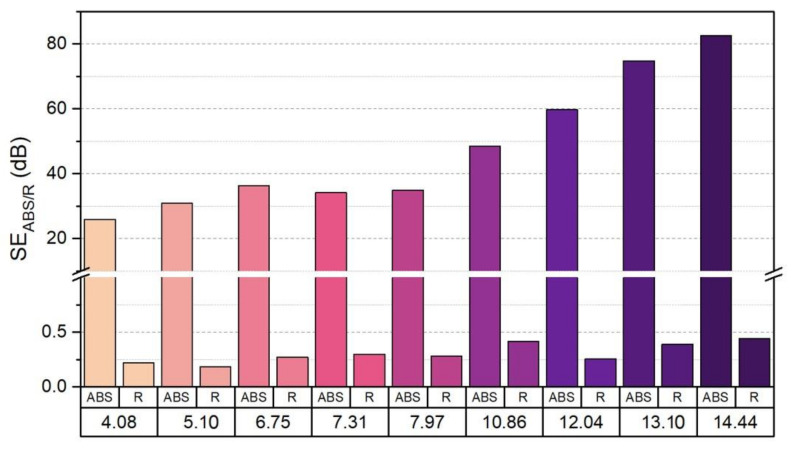
Comparison of the absorption and reflection components of total SE at 0.9 THz for samples with different CB loading.

**Figure 4 materials-14-00835-f004:**
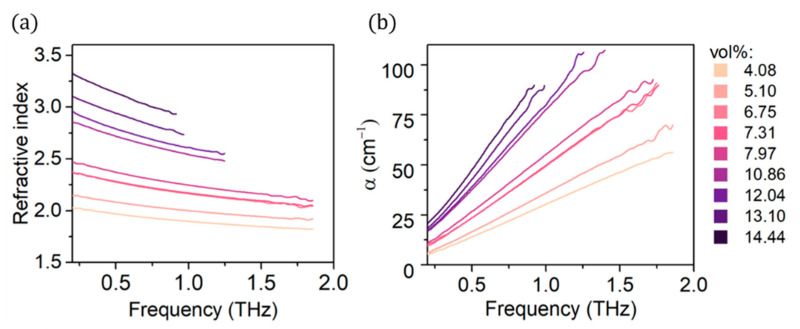
(**a**) Refractive index and (**b**) absorption coefficient of EBA/CB composites.

**Figure 5 materials-14-00835-f005:**
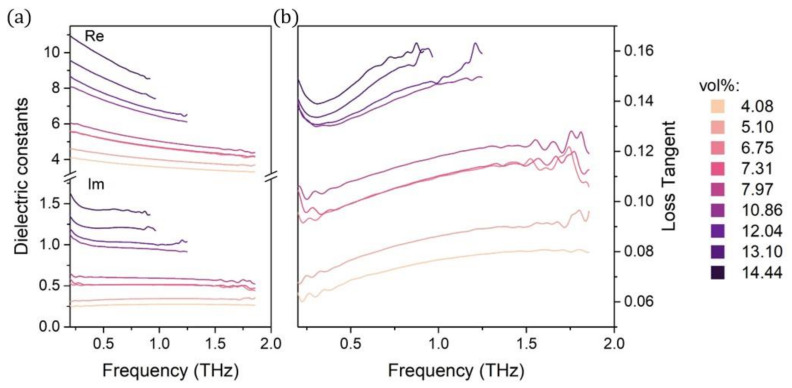
Dielectric properties of fabricated composites: (**a**) real and imaginary parts of the dielectric permittivity; and (**b**) the calculated loss tangent.

**Figure 6 materials-14-00835-f006:**
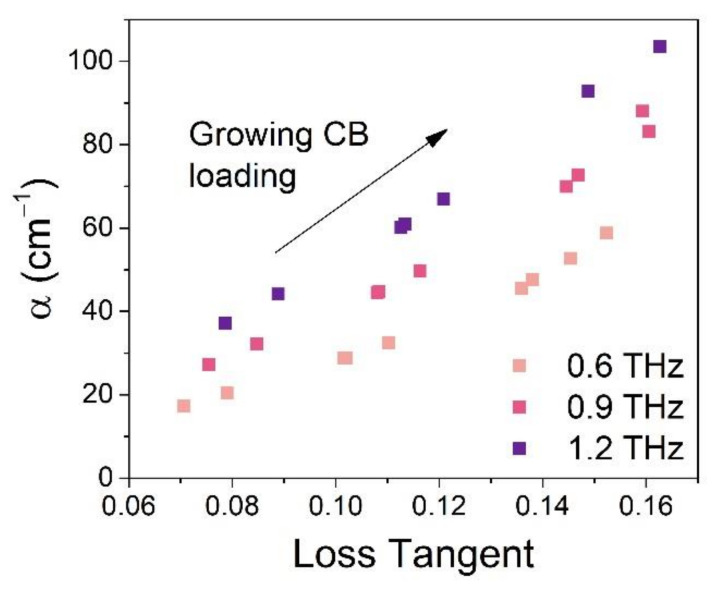
Relation between absorption coefficient and the loss tangent shown at three frequencies—0.6, 0.9 and 1.2 THz.

**Figure 7 materials-14-00835-f007:**
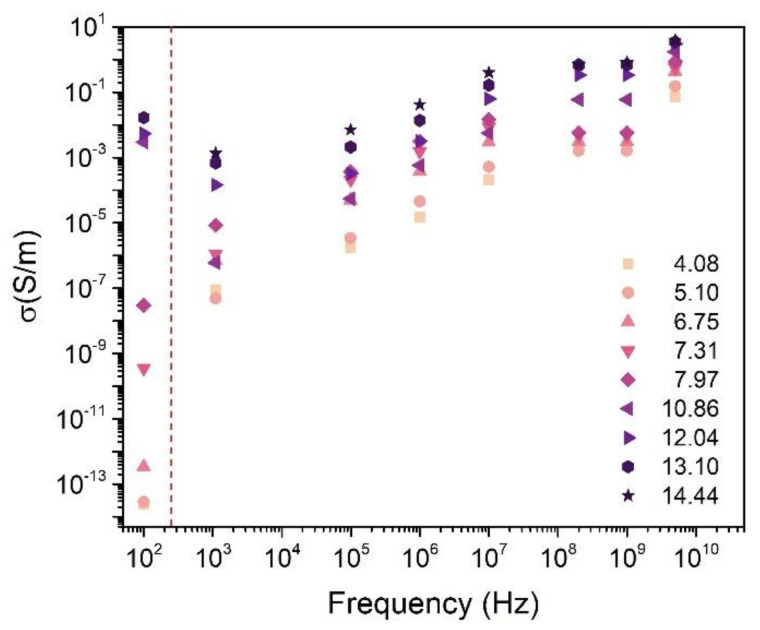
AC conductivity of CB composites as a function of frequency, additionally DC conductivity is presented.

## Data Availability

The raw data required to reproduce these findings are available on request from the corresponding author, K.Z.-C. The processed data required to reproduce these findings are available on request from the corresponding author, K.Z.-C.

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
