# Peer review of "Terahertz Shielding Properties of Carbon Black Based Polymer Nanocomposites"

_materials, 2021, doi:10.3390/ma14040835_

Round 1

Reviewer 1 Report

The manuscript involves investigation of a polymer (EBA) carbon black (CB) composite material as shielding materials to be used at terahertz frequency range. The experiments were designed neatly with proper investigative approaches and data were presented in a very comprehensive way. A sample with 14.44 vol% loading of CB with EBA is proven to be highly effective with a very high shielding efficiency (SE) value compared to its contenders with other types of carbon-based nanomaterials. I found the manuscript worthy of publishing in its current format with minor revisions.

Firstly, please include a rationale for your justification for stopping at 14.44 vol% loading of CB, as you have mentioned in the manuscript that problems due to agglomeration begins at 22 vol%. Why didn't you try any higher loading then?

Secondly, I believe the imaging part of your characterization needs to be improved if possible. You have included one representative SEM image of a composite material (didn't mention which sample was it). Please include a few images of samples with different loading to exhibit any issues with the agglomeration.

Finally, all of your graphs doesn't include the base values, that is the corresponding values of the pristine polymer. From readers perspective, it is always beneficial to have data of the baseline to get an idea of the improvement. I would suggest you include those data points in your graphs.

If the above mentioned issues are addressed properly, I believe this manuscript should be published in its current format.

Author Response

The manuscript involves investigation of a polymer (EBA) carbon black (CB) composite material as shielding materials to be used at terahertz frequency range. The experiments were designed neatly with proper investigative approaches and data were presented in a very comprehensive way. A sample with 14.44 vol% loading of CB with EBA is proven to be highly effective with a very high shielding efficiency (SE) value compared to its contenders with other types of carbon-based nanomaterials. I found the manuscript worthy of publishing in its current format with minor revisions.

Firstly, please include a rationale for your justification for stopping at 14.44 vol% loading of CB, as you have mentioned in the manuscript that problems due to agglomeration begins at 22 vol%. Why didn't you try any higher loading then?

A1. We would like to thank the reviewer for this question. The measurements via THz-TDS are limited by the signal strength and so the EMI SE can be measured up to a certain point. As one can see in Figure 2, we already had to cut the SETOT spectra at some point, because “the measurement became unreadable and unreliable” because the composite was so absorptive. Using higher CB loadings would not result in higher measured EMI shielding values, as that has been reached for all the composites at the level of ~80 dB, it would only impact measurable slope of SETOT (frequency). We have added the rationale to the revised manuscript (3.1 Shielding efficiency, line 178).

Secondly, I believe the imaging part of your characterization needs to be improved if possible. You have included one representative SEM image of a composite material (didn't mention which sample was it). Please include a few images of samples with different loading to exhibit any issues with the agglomeration.

A2. It is true, we have not mentioned which sample is presented in the SEM picture, it has been corrected in the reviewed manuscript. The SEM picture shows a cross-section of a 14.44 vol% loaded sample, which is the highest CB concentration used in this work. However the SEM was used only for the sake of visualisation. We added AFM-resiscope micrographs for three selected samples (Figure 1ain the revised manuscript). In Ref. 25 the SEM pictures have been given of highly loaded samples and shown a good dispersion at different length scales and CB contents.

Following paragraph was added in the revised manuscript (Subsection 2.1 Material EBA/CB, line 120):

‘The CB dispersion in the EBA matrix was investigated at nanometric scale by conductive probe atomic force microscopy (AFM) which when used in contact mode; it permits a mapping of the local electrical resistance simultaneously with classical topography images. The former information is more efficient to evaluate the effective dispersion of CB through the polymer matrix regarding the high difference of the electrical resistance between the two constituents. For illustration purposes, Figure 1a shows AFM images for three samples containing 6.75, 7.97 and 12.04 vol % of CB. This picture indicates clearly that the filler is finely dispersed in the polymer matrix. Complementary information was obtained by SEM as reported  elsewhere [25]’’

Finally, all of your graphs doesn't include the base values, that is the corresponding values of the pristine polymer. From readers perspective, it is always beneficial to have data of the baseline to get an idea of the improvement. I would suggest you include those data points in your graphs.

A3. Such information cannot be included in the manuscript since the fabricated samples are provided directly by Borealis AB (Sweden). Thus, we are unable to fabricate or perform measurements on pure EBA samples. However, based on the literature and our previous works we can state, the polymer matrix is mostly negligible in the case of EMI SE in terahertz range. To support this argument, we have prepared the following literature examples.

  • Zdrojek, Mariusz, et al. "Graphene-based plastic absorber for total sub-terahertz radiation shielding." Nanoscale28 (2018): 13426-13431.
  • Chamorro-Posada, Pedro, et al. "THz TDS study of several sp2 carbon materials: Graphite, needle coke and graphene oxides." Carbon98 (2016): 484-490.
  • Rungsawang, R., et al. "Terahertz spectroscopy of carbon nanotubes embedded in a deformable rubber." Journal of Applied Physics12 (2008): 123503.

If the above mentioned issues are addressed properly, I believe this manuscript should be published in its current format.

Reviewer 2 Report

This is a nice work, comprehensively studied the optical properties of polymer composites enhanced with various carbon black loading in terahertz range.

More characterisation are suggested to add in the manuscript to support the structure of the composites, such as TGA for showing the detailed ratios and SEM for all samples; Also, the title can be more specific, now is too broad; The authors are suggested to emphasize the novelty of the paper in the introduction part.

Author Response

This is a nice work, comprehensively studied the optical properties of polymer composites enhanced with various carbon black loading in terahertz range.

More characterisation are suggested to add in the manuscript to support the structure of the composites, such as TGA for showing the detailed ratios and SEM for all samples; Also, the title can be more specific, now is too broad; The authors are suggested to emphasize the novelty of the paper in the introduction part.

A1. Thank you for this suggestion. We note that the TGA experiments are already reported elsewhere [Ref 25] and show an increase of the thermal stability of composites with increasing CB amount. This result is partially related to the best dispersion of CB particles inside the matrix. In that Ref., the SEM pictures have been given of representative samples and shown a good dispersion at different length scales and CB content. Here, the atomic force microscopy (AFM) images were provided for selected samples confirming an homogenous dispersion of CB in the EBA matrix (see Figure 1a). In addition, the crystallinity of the EBA matrix (20%), measured by DSC techniques, increases with CB content where its value reaches ~30% for samples containing 14.4 vol.%  of CB particles as reported in [23].

We added these characteristics of our composites in the Subsection 2.1 Material EBA/CB, line 106.

Regarding the title. In our work we have presented a variety of EMI shielding and optical properties – the SETOT and its components, transmittance reflectance, absorption coefficient and refractive index, as well as some dielectric properties, namely the dielectric constants and loss tangent. We have also reported on the conductivity. As we put the most attention and value to the EMI shielding properties, it is reflected in the title of the manuscript. That is why we would like to keep the title as it is.

We have introduced changes in the introduction regarding novelty of the paper (1. Introduction, line 88 and 97).

Reviewer 3 Report

General comments

Work is ordinary but manuscript is filled with hyperboles and illogical claims. Needs major revision.

Specific comments

Authors never gave why they chose "ethylene co-butyl acrylate" when there are many cheaper, widely available, commodity rubbers like SBR, NBR, NR etc. Suitable justification for the choice of polymer matrix should be given in revised manuscript.

The statement "The CB dispersion state within the EBA matrix was investigated at nano-metric scale by conductive probe atomic force microscopy (AFM) as reported elsewhere [24, 25]." is misleading. Ref #24 : "Bentoumi M, Mdarhri A, Montagne A, et al (2019) Nano‐indentation for probing mechanical properties of nanocomposites based on ethylene butyl acrylate copolymer and carbon black. J Appl Polym Sci 136:47897." reports topology of only 8.4 vol% (Figure 2. A representative topographic (at left) and electrical (at right) AFM micrographs of sample containing 8.4 vol % of CB). But in this paper there is no 8.4 vol% as authors say in line # 122 (specifically 4.08, 5.10, 6.75, 7.31, 7.97, 10.86, 12.04, 13.10 and 14.44 vol%).  One can't extrapolate topology of 8.4 vol% CB to 10.86, 12.04, 13.10 and 14.44 vol%. There is a concept of percolation limit and above this limit extensive agglomeration is observed especially in acetylene blacks.

Additionally the acetylene black used in Ref #24 was "acetylene CB were obtained from Borealis AB (Sweden)" whereas in this study the acetylene CB was obtained from Denka which may or maynot have the same structure and characteristics.  This also raises serious questions about data presented in Fig. 7. (Figure 7 AC conductivity of CB composites as a function of frequency, additionally DC conductivity is presented.. Conductivity measurements for DC and frequencies from 1.1 kHz to 1 GHz reproduced from [25].)

Line 118 to 120: Remove the statement "Samples prepared for this study exhibited a uniform dispersion of the filler particles in the matrix, which allowed an enhancement of interfacial properties and optimum transfer of the mechanical stress between the filler and the matrix." No proof given.

There are many more problems with this manuscript but above are enough to recommend major revision.

Author Response

General comments

Work is ordinary but manuscript is filled with hyperboles and illogical claims. Needs major revision.

Specific comments

Authors never gave why they chose "ethylene co-butyl acrylate" when there are many cheaper, widely available, commodity rubbers like SBR, NBR, NR etc. Suitable justification for the choice of polymer matrix should be given in revised manuscript.

A1. We would like to thank the reviewer for this remark. Because of the complex family of each of these rubber polymers depending on their composition, i.e., the amount of BA in EBA, the acrylonitrile in NR and NBR, S/R ratio in SBR, a direct comparison seem to be difficult (Elastomers and Rubbers, 239-271 Chap 11 in: The Effect of Long Term Thermal Exposure on Plastics and Elastomers, edited by L.W. McKeen, Elsevier  (2014), https://doi.org/10.1016/B978-0-323-22108-5.00011-4,F. Findik et al. Materials and Design 25 (2004) 269–276. Roughly, high purity/crystallinity, narrower of molecular weight distribution (B-A. Sultan and E. Sorvik. J. Appl. Polym. Sci. 43,1761-1777 (1991)), thermal stability and aging behaviours(G. A. Schwartz et al. Polymer 44 7229–7240 (2003)) are some advantages of EBA compared to other ethylene copolymers and rubbers. In addition, the EBA polymer used here is compatible with CB type: easy processing, good dispersion and low processing cost with a low percolation threshold (~0.08) [24].

Justification for the choice of polymer matrix was added in section 2.1 Material: EBA/CB, line 113.

The statement "The CB dispersion state within the EBA matrix was investigated at nano-metric scale by conductive probe atomic force microscopy (AFM) as reported elsewhere [24, 25]." is misleading. Ref #24 : "Bentoumi M, Mdarhri A, Montagne A, et al (2019) Nano‐indentation for probing mechanical properties of nanocomposites based on ethylene butyl acrylate copolymer and carbon black. J Appl Polym Sci 136:47897." reports topology of only 8.4 vol% (Figure 2. A representative topographic (at left) and electrical (at right) AFM micrographs of sample containing 8.4 vol % of CB). But in this paper there is no 8.4 vol% as authors say in line # 122 (specifically 4.08, 5.10, 6.75, 7.31, 7.97, 10.86, 12.04, 13.10 and 14.44 vol%).  One can't extrapolate topology of 8.4 vol% CB to 10.86, 12.04, 13.10 and 14.44 vol%. There is a concept of percolation limit and above this limit extensive agglomeration is observed especially in acetylene blacks.

A2. We fully agree with the reviewer’s comment. This point is now addressed and the AFM micrographs were added (See  Fig. 1a) to evidence the good dispersion state of CB inside the matrix below and above the percolation threshold. In Ref [25], SEM pictures have been given for highly loaded samples that cannot be viewed by AFM (it is difficult to cut lamellae with few µm) and shown a good dispersion at different length scales and CB content.

Additionally the acetylene black used in Ref #24 was "acetylene CB were obtained from Borealis AB (Sweden)" whereas in this study the acetylene CB was obtained from Denka which may or may not have the same structure and characteristics.  This also raises serious questions about data presented in Fig. 7. (Figure 7 AC conductivity of CB composites as a function of frequency, additionally DC conductivity is presented.. Conductivity measurements for DC and frequencies from 1.1 kHz to 1 GHz reproduced from [25].)

A3. We precise that CB material is obtained from Denka Black and was used by Borealis AB to fabricate the EBA/CB composites.

Line 118 to 120: Remove the statement "Samples prepared for this study exhibited a uniform dispersion of the filler particles in the matrix, which allowed an enhancement of interfacial properties and optimum transfer of the mechanical stress between the filler and the matrix." No proof given.

A4. Please see A2.

There are many more problems with this manuscript but above are enough to recommend major revision.

Round 2

Reviewer 3 Report

Authors have made suitable corrections and made appropriate modifications in the revised manuscript. Accept.